# Quality of Life Assessment of Chronic Otitis Media Patients Following Surgery

**DOI:** 10.3390/jpm13010074

**Published:** 2022-12-29

**Authors:** Michele Cavaliere, Antonella Miriam Di Lullo, Pasquale Capriglione, Gaetano Motta, Elena Cantone

**Affiliations:** 1ENT Section, Department of Neuroscience, Reproductive and Odontostomatological Sciences, University of Naples, Federico II. Pansini Street n.5, 80131 Naples, Italy; 2Otorhinolaryngology, Head and Neck Surgery Unit, Department of Mental and Physical Health and Preventive, Medicine, Università Degli Studi Della Campania Luigi Vanvitelli, 80131 Naples, Italy

**Keywords:** COM, QoL, COMOT-15, chronic otitis media, ear surgery

## Abstract

Chronic otitis media (COM) is a persistent inflammation of the middle ear. COM often requires surgical management and represents one of the most disabling pathologies in the field of otolaryngology, not only due to hearing loss but also because recurrent otorrhea seriously affects the quality of life (QoL) of patients. The COMOT-15 questionnaire is a reliable, valid and sensitive tool for measuring the QoL of patients with COM. The aim of this study was to evaluate QoL by using the Italian version of the COMOT-15 in COM patients undergoing surgery based on age and different operation techniques. This observational retrospective study involved fifty-two consecutive patients undergoing surgical treatment for COM according to Nadol criteria. Preoperatively (T0) and 12 months after surgery (T1) patients underwent clinical examination, imaging, the Italian version of the COMOT-15 and pure tone audiometry. After surgery, we observed an improvement of QoL in 84.6% of the population. The COMOT-15 overall score, ear symptoms and hearing subscores showed significantly better ratings after surgery in the whole analyzed group. However, the separate analysis of patients operated with open techniques and closed techniques showed a significant improvement in ear symptoms subscore in both groups and a significant improvement in hearing subscore and mental health subscore only in patients operated on with closed techniques. Moreover, we observed a positive correlation between age and Δ-COMOT. This study shows the crucial role of a reliable and suitable questionnaire such as the COMOT-15 in evaluating COM patients, including clinical symptoms, functional and psychological impairments and highlighting a positive correlation between age and COMOT-15 results.

## 1. Introduction

Chronic otitis media (COM) is a persistent inflammation of the middle ear generally characterized by tympanic membrane perforation, ossicular chain impairment and bone resorption [1,2]. It seems to affect approximately 0.45–2.6% of the population [3], contributing over 50% to the global burden of the hearing loss [4]. 

COM often requires surgical management to avoid further disease progression and possible life-threatening complications such as facial nerve palsy, labyrinthitis, mastoiditis, meningitis, epidural, subdural and cerebral abscess formation [2]. 

Thus, COM represents one of the most disabling pathologies in the field of otolaryngology, not only due to hearing loss but also because recurrent otorrhea seriously affects the quality of life (QoL) of patients [5].

Recently, many researchers have published numerous articles on the impact of COM on daily life, and specific questionnaires have been proposed in the literature to define and evaluate the QoL of patients suffering from COM [5,6,7,8,9,10,11].

A recent work by Leilach et al. identified seven questionnaires, the Chronic Ear Survey (CES), Chronic Otitis Media-5 (COM-5), Chronic Otitis Media Outcome Test -15 (COMOT-15), Chronic Otitis Media Questionnaire 12 (COMQ 12), Chronic Otitis Media Benefit Inventory (COMBI), Zurich Chronic Middle Ear Inventory 21 (ZCMEI-21) and Stapesplasty Outcome Test 25 (SPOT-25), that specifically investigate the QoL of patients with COM and/or conductive hearing loss and evaluated the negative and positive features for each test [12]. According to the authors, the most reliable questionnaires are the COMOT-15, ZCMEI-21 and SPOT-25, although the SPOT-25 questionnaire is indicated for patients suffering from conductive hearing loss due to otosclerosis [13]. 

Recently, Mlinski et al. confirmed these data by identifying the COMOT-15 and ZCMEI-21, mostly used in German-speaking countries, as the most comprehensive questionnaires to investigate the clinical features of COM [14].

According to literature data, the COMOT-15 is a reliable, valid and sensitive tool for measuring the QoL of patients with COM, as demonstrated by its creator Baumann in 2009 [6] and confirmed by our recent work in which we validated the Italian version of the COMOT-15 [15].

The aim of this study was to evaluate QoL by using the Italian version of the COMOT-15 in COM patients undergoing surgery based on age and different operation techniques. 

## 2. Materials and Methods

This observational retrospective study approved by the Ethical Committee of the University of Naples “Federico II” was carried out in accordance with the Declaration of Helsinki as amended in 2014 [16]. Patients gave their written informed consent before starting data collection. A total of 52 consecutive patients, 28 females and 24 males (mean age 48.3 ± 16.1 SD), 38/52 (73%) affected by cholesteatoma, undergoing surgical treatment for COM according to Nadol criteria (disease of the middle ear or mastoid, or both, with irreversible mucosal change or infection lasting more than 3 months) were included in the study [7]. 

Inclusion criteria were patients aged 18 or above affected by monolateral COM (cholesteatomatous or not) lasting more than 6 months who underwent closed or open surgical procedures. We only considered patients with unilateral COM because a possible pathology on the other side (not undergoing surgery) could have altered the results obtained from surgery as regards the variation of QoL.

Exclusion criteria were age below 18, previous ear surgery or any other ear, nose, throat (ENT) diseases, previous diagnosis of severe chronic diseases or any other psychiatric diseases and/or loss of full legal capacity that reduce the patient’s compliance to answer the questionnaire. 

Clinical examination, imaging, pure tone audiometry and the QoL questionnaire were performed preoperatively (T0) and 12 months after surgery (T1). Clinical evaluation was conducted with the aid of the microscope. As regards imaging, a high-resolution CT was performed at T0 (before surgery) and a Multi-Shot non-EPI DWI-RM at T1 (1 year after). Pure-tone audiometry for testing conventional frequency range (0.125 to 8 kHz), was performed using an Amplaid 319 audiometer (Amplaid Inc., Milan, Italy) in a double-walled, soundproof room. Pure-tone average values were calculated in dB from the air conduction hearing loss as the mean of 0.5, 1, 2 and 4 kHz thresholds (pure tone average, PTA). 

To assess the QoL, all subjects were asked to answer the Italian version of the COMOT-15 (Figure 1).

The COMOT-15 consists of three subscales: (1) ear symptoms (ES), questions from 1 to 6; (2) hearing function (HF), questions from 7 to 9; and (3) mental health (MH) questions from 10 to 13. In addition to questions from 1 to 13, the COMOT-15 includes other 2 questions: a question on the general evaluation of COM’s impact on QoL (question 14), and a question indicating the frequency of ENT visits due to COM in the previous six months (question 15). The total score obtained by the sum of all items (1–15) and the subscores were transformed to a 0–100 scale by dividing the sum of the raw scores by 3 and multiplying by 4 [15].

For data analysis, we used the ∆-COMOT, i.e., the difference of COMOT values detected before (T0) and 12 months (T1) after surgery (positive ∆-COMOT = better QoL, negative ∆-COMOT = worse QoL). 

Surgery was performed on all patients by the same experienced surgeon (MC). Through a retroauricular incision, open techniques (31 canal wall down tympanoplasty (CWDT)) or closed procedures considered together (7 canal wall up tympanoplasty (CWUT), 12 underlay myringoplasty (U-MPL), and 2 overlay myringoplasty (O-MPL)) were performed according to the extension and the histological nature of the disease (cholesteatoma or not). Table 1 stratifies the type of surgery according to age. Under the same condition of the disease, we preferred to perform closed tympanoplasty in young patient, while myringoplasty (both under and over) was performed also in older people. Temporalis fascia was used for the reconstruction of the tympanic membrane in all cases. When indicated, we performed ossiculoplasty with incus remodeling (in 12/31 patients who underwent CWDT), titanium total (TORP—in 3/31 CWDT) and partial (PORP—in 4/31 CWDT and in 1/7 CWUT) ossicular replacement prostheses or tragal cartilage (in 5/31 CWDT and in 6/7 CWUT).

### Statistical Analysis

Statistical evaluation was carried out using SPSS v 13.0 for windows (SPSS Inc., Chicago, IL, USA). 

The significance of the differences between T0 and T1 was evaluated by Student’s *t* test. Connections between variables were calculated with Pearson’s correlation coefficient. The significance level for all tests was set at *p* < 0.05.

## 3. Results

After the surgical procedures we observed an improvement of the clinical and functional data and also of the QoL of the operated patients. In particular, there was no recurrence of cholesteatoma at T1 in all cases. To eliminate otorrhea in only 6/52 patients (4 CWDT, 1 CWUT, 0 U-MPL, 1 O-MPL) we prolonged the use of ear drops as compared to the usual follow-up medications (on average one month). Less brilliant results were obtained in the objective evaluation of hearing. At T0 auditory PTA was 55 dB in the overall population (PTA was 62 dB in patients operated with the open technique and 45 dB in those subjected to the closed technique). At T1, the means had an improvement but not an excellent one (PTA in the general population was 35 dB; PTA was 45 dB and 20 dB in the open and closed techniques, respectively). 

As for the QoL, an improvement occurred in 84.6% of the population (44 patients: 25 CWDT, 6 CWUT, 11 U-MPL, 2 O-MPL); a worsening of QoL occurred in 15.4% (8 patients: 6 CWDT, 1 CWUT, 1 U-MPL).

The COMOT-15 overall score (OS) (OS:T0: 36.76 ± 13.1 vs. T1: 26.88 ± 12; ∆-COMOT = +9.88, *p*: 0.00011) and ES and HF subscores (ES: T0: 11.05 ± 4.45 vs. T1: 4.15 ± 3.61, ∆-COMOT = +6.9, *p*: 0.000007; HF: T0: 13.48 ± 5.83 vs. T1: 12.17 ± 5.71, ∆-COMOT = +1.31, *p*: 0.02511) showed significantly better ratings after surgery. We did not find statistically significant improvement after surgery for the MH subscore (T0: 9.69 ± 4.65 vs. T1: 7.903 ± 5.409, ∆-COMOT = +1.79, *p*: 0.073) (Figure 2). 

We observed a positive direct linear correlation (r = 0.159438243; *p* = 0.0259) between age and ∆-COMOT. Older age seems to be correlated with better QoL improvement (Figure 3).

In open techniques, we observed a significant improvement in OS after surgery (T0: 39.6 ± 13.4 vs. T1: 30.29 ± 10.85; ∆-COMOT = +9.31 *p*: 0.000072) due mostly to ES reduction subscores (T0: 12.03 ± 4.16 vs. T1: 4.54 ± 4.20; ∆-COMOT = +7.49, *p*: 0.0000837), whereas we did not observe significant variation in the HF (T0: 13.9 ± 6.26 vs. T1: 13.78 ± 4.95; ∆-COMOT = +0.12, *p*: 0.68) and MH items (T0: 10.41 ± 4.85 vs. T1: 9.38 ± 5.48; ∆-COMOT = +1.03, *p*: 0.20) (Figure 4).

However, in closed techniques, the patients showed a significative improvement of QoL (T0: 33.23 ± 12.05 vs. T1: 21.85 ± 11.92; ∆-COMOT = +11.38, *p*: 0.000051) not only due to the reduction of ES subscores (T0: 9.61 ± 4.57 vs. T1: 3.57 ± 2.42; ∆-COMOT = +6.04, *p*: 0.0000031) but also to the reduction of HF (T0: 12.85 ± 5.21 vs. T1: 10.14 ± 6.26; ∆-COMOT = +2.71, *p*: 0.038) and MH subscores (T0: 8.61 ± 4.23 vs. T1: 5.71 ± 4.58; ∆-COMOT = +2.90, *p*: 0.00015) (Figure 5). 

The obtained results are summarized in Table 2.

## 4. Discussion

According to the literature, surgery represents the most effective therapy for COM patients to prevent further worsening of the disease and possible life-threating complications. Especially the cholesteatoma must be operated (there are no conservative treatment options) to avoid bone destruction and dangerous intracranial and intratemporal problems [2,17]. 

So far, medical therapy is generally prescribed preoperatively or when surgery is contraindicated [17,18,19,20]. Although surgery is a common event in the life of a patient with COM, it is very interesting to investigate all benefits that it will bring to patients’ QoL even if the improvement of QoL after cholesteatoma surgery is important, but it should not be a factor that limits the decision about performing surgery.

The availability of several specific questionnaires for the assessment of QoL in patients with COM has prompted many authors to address the topic. Bächinger et al., comparing the score of the ZCMEI-21 questionnaire before and after surgery, calculated a minimal clinically important difference (MCID) = 5.3 (SD 12.0) above which the intervention can be considered satisfactory for the patient [21]. 

Other authors have analyzed the impact of preoperative comorbidities on surgical outcomes. Leilach et al. identified depressed individuals as the worst patients to undergo surgery [3], whereas Choi et al. identified diabetic patients as those with a higher incidence of postoperative complications, probably due to higher difficulty in wound healing and re-epithelialization of the surgical cavity [22]. Another interesting finding from this study is that the authors described worse scores on the QoL questionnaire (The Korean version of the Chronic Ear Survey, K-CES) in subjects with high socioeconomic level [22]. Eventually, we correlated the subjects’ age with QoL and found that young patients reported worse COMOT-15 scores after surgery (Figure 3), possibly because they have higher life expectations than older patients, and, in addition, they are less prone to accept hearing impairments (because of difficulties in communication), and COM-induced handicap that restricts their daily activities [2].

Most of the previous publications have focused on QoL based on surgical techniques and/or on the severity of COM (presence or absence of cholesteatoma, extension of the inflammatory process, ossicular chain status, etc.) [18,19,20]. These variables have linked each other. It is clear, in fact, that patients with severe COM should undergo a more invasive surgery. However, that does not seem to affect long-term postoperative QoL. In the study by Maile et al., there are no statistically significant differences in QoL based on the presence or absence of cholesteatoma, although they used a generic questionnaire such as the Glasgow Benefit Inventory (GBI) and not one disease-specific such as the COMOT-15 [23]. 

Weiss et al., using the ZCMEI-21 questionnaire, suggest that the presence of cholesteatoma is not decisive for the assessment of QoL, but the real factor influencing postoperative QoL is the air conduction threshold [24].

In another work, the same authors stressed the importance of mastoid cavity reconstruction with obliterative techniques to improve patients’ QoL [25]. According to Lucidi et al., by using the CES questionnaire, there are no statistically significant differences, especially in the long term, between patients undergoing closed tympanoplasty compared to those undergoing open tympanoplasty [26].

In our experience, most patients improved their QoL after surgery independently for the presence or absence of cholesteatoma according to previous data [23,24]. This is the reason why we divided our sample according only to surgical procedure, considering only patients with unilateral COM, to avoid the fact that the outcomes can be influenced by the pathology of the contralateral side. 

In our study population, the COMOT-15 OS and the ES and HF subscores showed significant improvement after surgery. These results may be related to the reduction of most of the sequelae of COM, such as hearing loss, tinnitus, dizziness, otorrhea, as confirmed by many authors [2]. In fact, regarding hearing loss, we obtained an improvement of an average of 20 dB in the overall population (55 dB at T0 vs. 35 dB at T1). This improvement was slightly better in closed techniques (45 dB at T0 vs. 20 dB at T1) than the open techniques (62 dB at T0 vs. 45 dB at T1).

Considering the type of surgical techniques (open or closed), our study revealed a significant improvement in OSafter open techniques, mainly due to the reduction of ES subscores, whereas we did not observe significant variation in the HF and MH items. In the closed techniques, we found a significant improvement in QoL not only for the reduction of ES subscores but also for the reduction of HF and MH subscores. These findings can be interpreted from two different points of view. On one hand, our data confirmed a good sensitivity of the COMOT-15 in the analysis of auditory results, as also stated by Baumann, who found a close correlation between the data obtained by the COMOT-15 and those objectively obtained from preoperative and postoperative audiometric thresholds [27].

On the other hand, it can be hypothesized that less invasive surgery, such as closed ones, ensures higher patient satisfaction due to better functional results [18,19,20].

The latter datum (i.e., the greater satisfaction found in closed techniques) may appear in contrast with the other datum (which highlights a greater postoperative satisfaction in older patients Figure 2) since closed tympanoplasty has been preferentially performed in young patients, as reported in Table 1. 

However, this apparent contradiction can be explained by taking into account two elements:

(1) The closed techniques included not only the 7 CWUT but also 14 MPL in which older patients were also represented.

(2) Older patients generally started from a more complicated preoperative condition for which even partial results led to a good degree of satisfaction.

The limitations of this research are the monocentric retrospective observational design of the study, small sample size and the exclusion of patients who underwent revision surgery [23]. 

## 5. Conclusions

This study highlights the crucial role of a reliable and suitable questionnaire such as the COMOT-15 to evaluate patients affected by COM, including clinical symptoms (ES) and functional (HF) and psychological impairments (MH). 

Through the COMOT-15, we have revealed significant improvements of total COMOT-15 scores in patients who underwent surgery of COM both with the open and closed techniques. In both groups, the improvement was caused by improvement in the ear symptoms score, but in cases of patients who underwent closed techniques, improvement was also due to better functional results and, thus, emotional ones. 

In addition, we note a positive correlation between age and improvement in COMOT-15 score after surgery. 

## Figures and Tables

**Figure 1 jpm-13-00074-f001:**
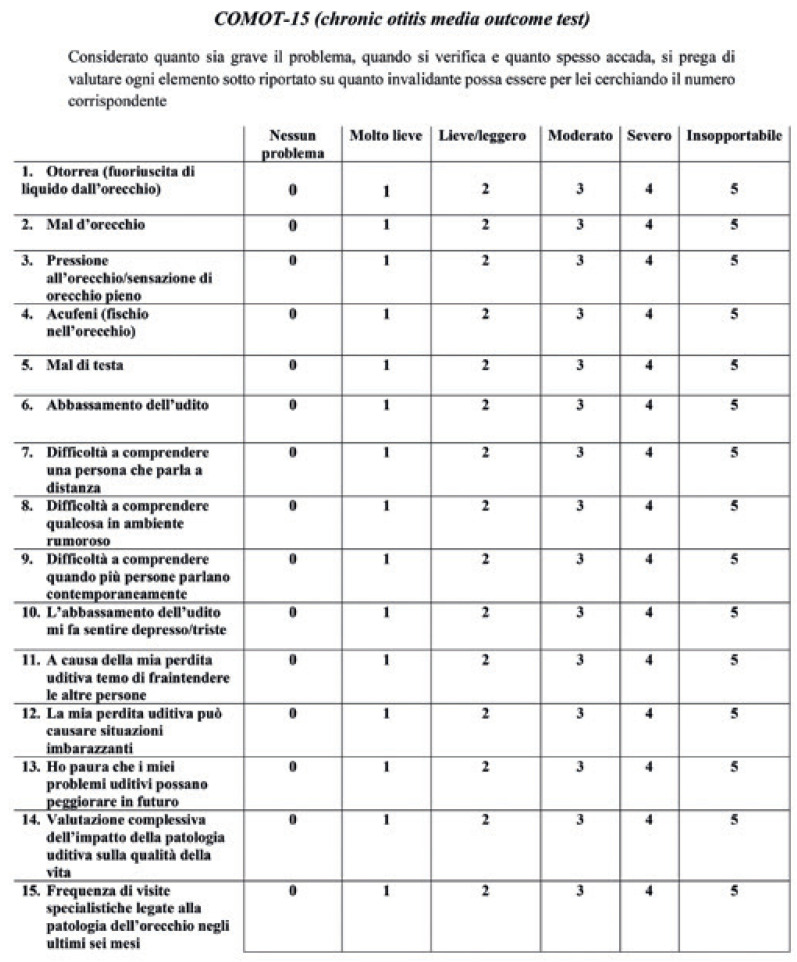
The Italian and English versions of the COMOT-15.

**Figure 2 jpm-13-00074-f002:**
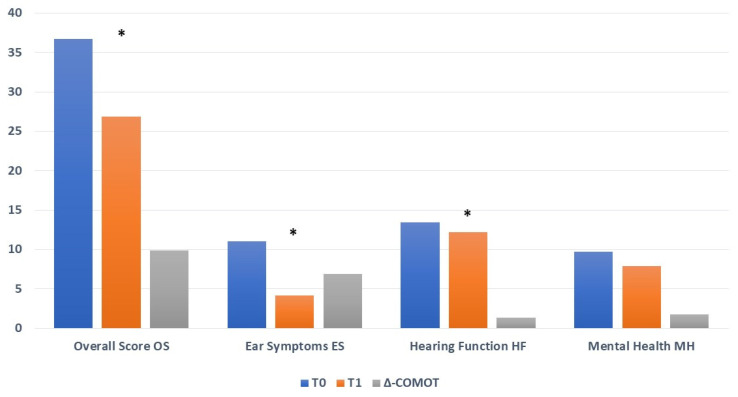
COMOT-15 values before surgery (T0) and twelve months later (T1); * statistically significant.

**Figure 3 jpm-13-00074-f003:**
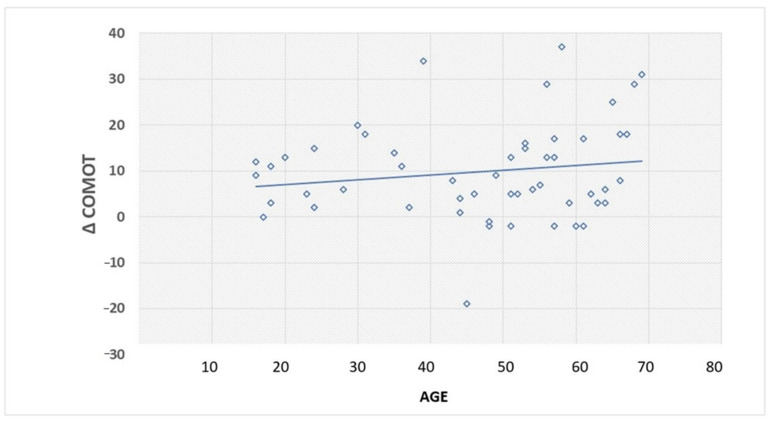
Correlation between age and ∆-COMOT.

**Figure 4 jpm-13-00074-f004:**
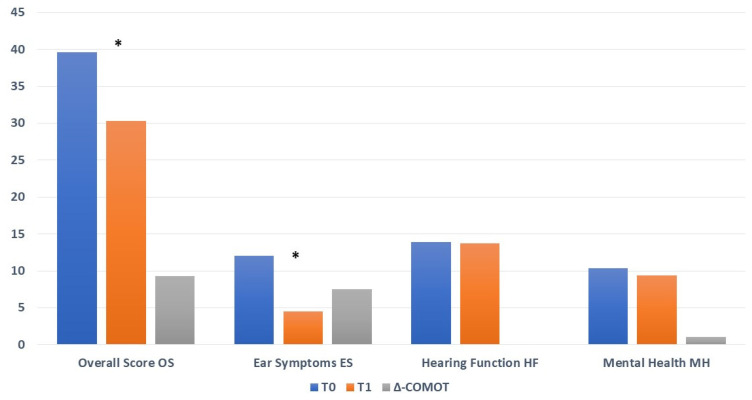
COMOT values for open techniques before surgery (T0) and twelve months later (T1); * statistically significant.

**Figure 5 jpm-13-00074-f005:**
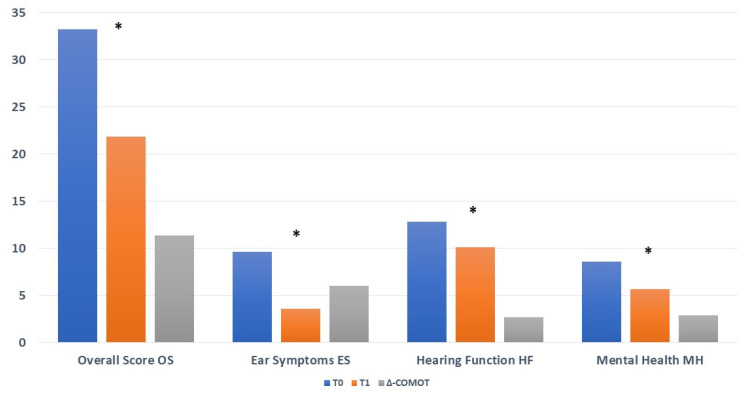
COMOT values for closed techniques before surgery (T0) and twelve months later (T1); * statistically significant.

**Table 1 jpm-13-00074-t001:** Stratification of surgery by age.

Age (Years)		≤50	>50	Total
Closed Techniques	CWUT	6	1	7
	U-MPL	7	5	12
	O-MPL	1	1	2
Open Techniques	CWDT	10	21	31
Total		24	28	52

**Table 2 jpm-13-00074-t002:** The results of COMOT-15 and its subscales for the whole group and analyzed subgroups separately.

	Open Techniques	Closed Techniques		
Type of intervention	CWDT	CWUT	U-MPL	O-MPL
Patients (nr).	31	7	12	2
QoL OS T0	36.76 ± 13.1			
QoL OS T1	26.88 ± 12			
∆-COMOT OS	+9.88 (*p*: 0.00011) *			
QoL ES T0	11.05 ± 4.45			
QoL ES T1	4.15 ± 3.61			
∆-COMOT ES	+6.9 (*p*: 0.000007) *			
QoL HF T0	13.48 ± 5.83			
QoL HF T1	12.17 ± 5.71			
∆-COMOT HF	+1.31 (*p*: 0.02511) *			
QoL MH T0	9.69 ± 4.65			
QoL MH T1	7.903 ± 5.409			
∆-COMOT MH	+1.79 (*p*: 0.073)			
QoL OS T0	39.6 ± 13.4	33.23 ± 12.05		
QoL OS T1	30.29 ± 10.85	21.85 ± 11.92		
∆-COMOT OS	+9.31(*p*: 0.000072) *	+11.38 (*p*: 0.000051) *		
QoL ES T0	12.03 ± 4,16	9.61 ± 4.57		
QoL ES T1	4.54 ± 4.20	3.57 ± 2.42		
∆-COMOT ES	+7.49 (*p*: 0.0000837) *	+6.04 (*p*: 0.0000031) *		
QoL HF T0	13.9 ± 6.26	12.85 ± 5.21		
QoL HF T1	13.78 ± 4.95	10.14 ± 6.26		
∆-COMOT HF	+0.12 (*p*: 0.68)	+2.71 (*p*: 0.038) *		
QoL MH T0	10.41 ± 4.85	8.61 ± 4.23		
QoL MH T1	9.38 ± 5.48	5.71 ± 4.58		
∆-COMOT MH	+1.03 (*p*: 0.20)	+2.90 (*p*: 0.00015) *		

* statistically significant.

## Data Availability

The data presented in this study are available on request from the corresponding author. The data are not publicly available due to privacy.

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
