# Peer review of "Quality of Life Assessment of Chronic Otitis Media Patients Following Surgery"

_jpm, 2022, doi:10.3390/jpm13010074_

Round 1
Reviewer 1 Report
The article “Quality of life assessment of chronic otitis media patients following surgery” deals with the important topic of the quality of life perceived by patients suffering from chronic otitis media.
In the introduction, the reader is introduced to the studied issues. The basic terminological concepts are explained, the characteristics of the research problem are presented and the purpose of the work is clearly defined.
The material and research methods are substantively characterized.
The authors present the results of their research both descriptively and graphically in the form of figures. Statistical significance was given for each calculated index.
In the discussion, the authors referred to the results published by other authors, which allows the reader to read other opinions and have a broader view of the issue under discussion.
Major comments:
In my opinion the important limitation of the study is a small research group. Especially the group of patients operated with CWU technique (7 cases) should be bigger to have any binding conclusions. What is more, I think that the results of patients after myringoplasty (U-MPL and O-MPL) should be also presented in the manuscript. However, I’m not sure which cases are presented in group “Closed techniques” – CWUT only or CWUT and U-MPL and O-MPL together? It should be clarified in the manuscript. Define precisely what do mean: open techniques and close techniques.
In Materials and Methods the authors wrote, that: “Clinical examination, imaging, QoL questionnaire and pure tone audiometry were performed pre-operatively (T0) and 12 months after surgery (T1)” but in the Results section there are only information about QoL results, but there is nothing written about otological and audiological conditions before and after surgery (has it really improved?). In my opinion such a correlation between QoL and change in otological and audiological state of the patients would be very beneficial. We don’t know how many patient from the analyzed group have: recurrence of cholesteatoma after one year, still discharge from the ear instead of surgery and hearing improvement. All these parameters can affect the QoL after surgery. The authors wrote, that “In open techniques … we did not observe significant variation in the HF (T0: 13,9 ± 6.26 vs T1: 13,78 129 ± 4.95; ∆-COMOT=+0.12, p: 0,68)…” but they don’t present the audiological results. In the Discussion the authors wrote “ … , our data confirmed a good sensitivity of COMOT-15 in the analysis of auditory results,…” but they don’t present any auditory results.
The another week point of the study is poor defined surgical procedure. For example the authors give us only the information about the main type of surgery (CWDT, CWUT, myringoplasty), but without any details. For example there is no information about hearing reconstruction procedures in CWUT and CWDT patients. Even if you have open cavity you can also do myringoplasty and ossiculoplasty, and create a small tympanic cavity with the reconstructed tympanic membrane covering the head of stapes or short prosthesis, which can give quite good hearing results.
The important point of the study is that authors observed “a positive direct linear correlation (r = 0.159438243; p = 0.0259) between age and ∆ COMOT. Older age seems to be correlated with better QoL improvement”. On the other hand the authors obtained better results for CWU (∆-COMOT=+11.38) than CWD (∆-COMOT=+9.31). Could the authors explain what type of surgery (CWD/CWU) was preferred/done in older and in younger patients. Usually in young patients CWU is much more preferred than CWD, and in older patients CWU is still preferred , but CWD is also acceptable (especially in elderly patients with comorbidities).
Minor comments:
Summary:
Line 28: remove “mastoid cavity”, because mastoid cavity is anatomical part of the middle ear. It should be: “Chronic otitis media (COM) is a persistent inflammation of the middle ear”.
Line 34 – change “involving” into “involved”
Introduction:
Line 46 - remove “mastoid cavity”, because mastoid cavity is anatomical part of the middle ear. It should be: “Chronic otitis media (COM) is a persistent inflammation of the middle ear,…”.
Line 61 – why do you write that Stapesplasty Outcome Test 25 (SPOT-25) is one of the tests that specifically investigate the QoL of patients with COM? The title of the study by Leilach et al. is “State of the art of quality-of-life measurement in patients with chronic otitis media and conductive hearing loss”. In line 64 you write that SPOT-25 questionnaire is indicated for patients suffering from otosclerosis. Please clarify it.
Material and Methods:
Line 77 – explain Nadol criteria
Line 83 – explain what type of imaging was performed before surgery (CT?) and 1 year after (MRI DWI NON-EPI?)
Lines 89-91 – the sentence “. The total score obtained by the sum of all items (1-15) and the subscores were transformed in a 0-100 scale by dividing the sum of the raw scores by the sum of spans of the items multiplying by 100” is not clear for me. Could you write it more clear?
Line 95 – change “-“ into “[“ before 31 “closed techniques ̶31 canal wall down” into “closed techniques [31 canal wall down”
Results:
Line 110 – in my opinion Fig. 1 is unnecessary. The same data are written in the main text (lines 107-108).
Lines 113-116 – improve like this:
“The COMOT-15 overall score (OS) (OS:T0: 36.76 ± 13.1 vs T1: 26.88 ± 12; ∆-COMOT=+9.88, p:0.00011) and ES and HF subscores (ES: T0:11.05 ± 4.45 vs T1:4.15 ± 3.61, ∆-COMOT= +6.9, p:0.000007; HF: T0:13.48 ± 5.83 vs T1:12.17 ± 5.71, ∆-COMOT=+1.31, p: 0.02511) showed significantly better ratings after surgery. We did not find statistically significant improvement after surgery for MH subscore (T0:9.69 ± 4.65 vs T1:7.903 ± 5.409, ∆-COMOT=+1.79, p:0.073)”
Line 113-117 (main text and Figure 2) – could you explain why the difference for HF +1,31 is statistically significant and the difference +1,79 for MH is statistically not significant? As I understand the number of cases for both subscores is the same.
Line 123 (Figure 3) – add age in numbers (years) at the horizontal axis.
Line 132 Change into: “COMOT values for open techniques before surgery (T0) and twelve months later (T1); (*statistically significant).” It is important that these results are for open techniques so this information should not be in the parentheses.
Line 138: add + before 2.90 in the sentence: “..MH subscores (T0: 8.61 ± 4.23 vs T1: 5.71 ± 4.58; ∆-COMOT=+2.90, p:0.00015)..”
Line 140 Change into: “COMOT values for closed techniques before surgery (T0) and twelve months later (T1); (*statistically significant).” It is important that these results are for closed techniques so this information should not be in the parentheses.
Discussion:
Add few sentences about age and preferred technique and obtained results (see major comments).
Add the information that: 1) cholesteatoma is a disease that must be operated to avoid bone destruction and dangerous intracranial and intratemporal complications, 2) there are no conservative treatment options for cholesteatoma, 3) the improvement of QoL after cholesteatoma surgery is important, but it should not be a factor which limits the decision about performing surgery.
Line 177 – change “deafness” into “hearing loss”
Conclusions:
Lines 194-196: Remove sentences “Nowadays it can be considered the only questionnaire that evaluates the frequency of ENT visits. Moreover, it represents a quick and easy subjective way of investigation that reflects QoL much more closely than exclusive clinician- rated outcomes.”, because they are not conclusions from the current study.
Lines 197-199: Change sentence into: “Through COMOT-15, we have revealed significant improvements of total COMOT-15 scores in patients underwent surgery of COM both with the open and closed techniques. In both groups the improvement was caused by improvement in ear symptoms score, but in cases who underwent closed techniques also due to better functional results and so emotional ones”.
Lines 200-201: Change sentence into: “In addition, we note a positive correlation between age and improvement in COMOT-15 score after surgery.”
Author Response
Point-by-point responses |
Page |
Paragraph |
Line details |
|
1. Define the mean of open and closed techniques |
4 |
Material and methods |
104-107 |
I have clarified the mean of open and closed techniques as follow: “Through a retroauricular incision, open techniques [31 canal wall down tympanoplasty (CWDT)] or closed procedures considered together [7 canal wall up tympanoplasty (CWUT), 12 underlay myringoplasty (U-MPL), and 2 overlay myringoplasty (O-MPL)] were performed according to the extension and the histological nature of the disease (cholesteatoma or not)..” |
2. Otological and audiological results |
4 |
Results
|
120-127 |
I have clarified the audiological results: “After the surgical procedures we observed an improvement of the clinical, functional data and also of the QoL of the operative patients. In particular, there was no recurrence of cholesteatomatous pathology at T1 in all cases. To eliminate otorrhea in only 6/52 patients (4 CWDT, 1 CWUT, 0 U-MPL, 1 O-MPL) we had to prolong the use of ear drops for a longer period than that usually foreseen by the usual follow-up medications (on average one month). Less brilliant results were obtained in the objective evaluation of hearing. Pure tone audiometry showed at T0 an auditory PTA was 55 dB in the overall population (PTA was 62 dB in patients operated with the open technique and 45 dB in those subjected to the closed technique). At T1 the means have had an improvement but not excellent (PTA in the general population was 35 dB; PTA was 45 dB and 20 dB in the open and closed techniques, respectively).” |
3. Define surgical procedures |
4 |
Material and methods
|
104-112 |
Through a retroauricular incision, open techniques [31 canal wall down tympanoplasty (CWDT)] or closed procedures considered together [7 canal wall up tympanoplasty (CWUT), 12 underlay myringoplasty (U-MPL), and 2 overlay myringoplasty (O-MPL)] were performed according to the extension and the histological nature of the disease (cholesteatoma or not). Under the same condition of the disease, we preferred to perform closed tympanoplasty in young patient while myringoplasty (both under and over) was performed also in older people. Temporalis fascia was used for the reconstruction of the tympanic membrane in all cases. When indicated, we performed ossiculoplasty with incus remodeled (in 12/31 patients underwent to CWDT and in 3/7 CWUT) , titanium total (TORP – in 3/31 CWDT) and partial (PORP - in 4/31 CWDT and in 1/7 CWUT) ossicular replacement prostheses or tragal cartilage (in 5/31 CWDT and in 6/7 CWUT). |
4. Explain correlation between age and Δ Comot |
10 |
Discussion |
220-227 |
The latter datum (i.e. the greater satisfaction found in closed techniques) may appear in contrast with the other datum (which highlights a greater post-operative satisfaction in older patients- fig. 2) since closed tympanoplasty has been preferentially performed in young patients. However, this apparent contradiction can be explained by taking into account two elements: 1) the closed techniques included not only the 7 CWUT but also 19 MPL in which older patients were also represented. 2) older patients generally started from a more complicated pre-operative condition for which even partial results led to a good degree of satisfaction.
|
1) Reviewer 1 - Minor Comments
Point-by-point responses |
Page |
Paragraph |
Line details |
|
1.Remove mastoid cavity |
1 |
Summary |
28 |
Chronic otitis media (COM) is a persistent inflammation of the middle ear |
2. Change involving with involved |
2 |
Summary |
34 |
This observational retrospective study involved fifty-two consecutive patients |
3. Remove mastoid cavity |
2 |
Introduction |
46 |
Chronic otitis media (COM) is a persistent inflammation of the middle ear.. |
4. Clarify SPOT-25 ?? |
2 |
Introduction |
61-62 |
A recent work by Leilach et al. identified seven questionnaires ̶ Chronic Ear Survey (CES), Chronic Otitis Media-5 (COM-5), Chronic Otitis Media Outcome Test -15 (COMOT-15), Chronic Otitis Media Questionnaire 12 (COMQ 12), Chronic Otitis Media Benefit Inventory (COMBI), Zurich Chronic Middle Ear Inventory 21 (ZCMEI-21), Stapesplasty Outcome Test 25 (SPOT-25) ̶ that specifically investigate the QoL of patients with COM and/or conductive hearing loss by evaluating the negative and positive features for each test [12] . According to the authors, the most reliable questionnaires are COMOT-15, ZCMEI-21 and SPOT-25, although the SPOT-25 questionnaire is indicated for patients suffering from conductive hearing loss due to otosclerosis [13]. |
5. Explain Nadol Criteria |
2-3 |
Material and Methods |
77 |
…according to Nadol criteria (disease of the middle ear or mastoid, or both, with irreversible mucosal change or infection lasting more than 3 months) were included in the study. |
6. Explain what type of imaging was performed before surgery and 1 year after |
3 |
Material and Methods |
83 86-87-88 |
As regards imaging, a high resolution CT was performed at T0 (before surgery) and a Multi-Shot non-EPI DWI-RM at T1 (1 year after) |
7. The total score obtained by the sum of all items (1-15) and the subscores were transformed in a 0-100 scale…. Clarify ??? |
4 |
Material and Methods |
89-91 |
The total score obtained by the sum of all items (1-15) and the subscores were transformed in a 0-100 scale by dividing the sum of the raw scores by 3 and multiplying by 4. |
8. Change.. |
4 |
Material and Methods |
95 |
Through a retroauricular incision, open techniques [31 canal wall down tympanoplasty (CWDT)] or closed procedures considered together [7 canal wall up tympanoplasty (CWUT), 12 underlay myringoplasty (U-MPL), and 2 overlay myringoplasty (O-MPL)] |
9. Delate Fig.1 |
4 |
Results |
110 |
Delated Fig.1 |
10. Improve like this … |
4 |
Results |
113-116 |
The COMOT-15 overall score (OS) (OS:T0: 36.76 ± 13.1 vs T1: 26.88 ± 12; ∆-COMOT=+9.88, p:0.00011) and ES and HF subscores (ES: T0:11.05 ± 4.45 vs T1:4.15 ± 3.61, ∆-COMOT= +6.9, p:0.000007; HF: T0:13.48 ± 5.83 vs T1:12.17 ± 5.71, ∆-COMOT=+1.31, p: 0.02511) showed significantly better ratings after surgery. We did not find statistically significant improvement after surgery for MH subscore (T0:9.69 ± 4.65 vs T1:7.903 ± 5.409, ∆-COMOT=+1.79, p:0.073)” (Fig.2).
|
11. Main text and figure 2 |
4 |
Results |
113-117 |
In fact the number of cases for both subscores is the same but the number of items for HF and MH are different. |
12. Modify Figure 3 |
5 |
Results |
123 |
Modified fig.3 |
13. Change into … |
6 |
Results |
132 |
COMOT values for open techniques before surgery (T0) and twelve months later (T1); *statistically significant.
|
14. Add + before 2.90 |
6 |
Results |
138 |
MH subscores (T0: 8.61 ± 4.23 vs T1: 5.71 ± 4.58; ∆-COMOT= +2.90, p:0.00015 |
15. Change into … |
7 |
Results |
140 |
COMOT values for closed techniques before surgery (T0) and twelve months later (T1); *statistically significant |
16. Add sentences about age and preferred techniques |
9 |
Discussion |
221-227 |
The latter datum (i.e. the greater satisfaction found in closed techniques) may appear in contrast with the other datum (which highlights a greater post-operative satisfaction in older patients- fig. 2) since closed tympanoplasty has been preferentially performed in young patients. However, this apparent contradiction can be explained by taking into account two elements: 1) the closed techniques included not only the 7 CWUT but also 19 MPL in which older patients were also represented. 2) older patients generally started from a more complicated pre-operative condition for which even partial results led to a good degree of satisfaction.
|
17. Change “deafness” into “hearing loss” |
9 |
Discussion |
177 |
Changed |
18. Remove sentences |
9 |
Conclusions |
194-196 |
Removed |
19. Change sentences into .. |
9 |
Conclusions |
197-199 |
Changed |
20. Change sentence into… |
9 |
Conclusions |
200-201 |
Changed |

Reviewer 2 Report
Dear Authors
The Manuscript is interesting, the topic is relevant.However, it will be necessary to revise some aspects.Follow my suggestions:
Material and Methods: Only patients with unilateral COM were included.Why? I suggest that this important data be considered in the discussion and conclusion. Could it be included in the title? The Italian COMOT-15 Version could be included in the article. It is a recent material. It is important to include a more detailed description of the applied procedures.
Results and discussion: It is important to present descritives data of the sample.I suggest presenting the data in tabel with the respective statistical result. Audiological evaluation results could be presented and correlated with the COMOT-15 data. Why not study the group of patients with cholesteatoma in this study?Were the 14 and 15 questions analyzed?
Author Response
1) Reviewer 2
1. only patients with unilateral COM were included… |
3
9 |
Material and Methods
Discussion |
79-91
207-208 |
So, this is the reason why we divided our sample according only to surgical procedure, considering only patients with unilateral COM to avoid that the outcomes can be influenced by the pathology of the contralateral side.
… considering only patients with unilateral COM to avoid that the outcomes can be influenced by the pathology of the contralateral side. |
2. the Italian COMOT 15 version could be included in the study.. |
3 |
Material and Methods |
93 |
We have added Fig. 1 : The italian version of COMOT-15 . |
3. Presentation of the data in table with the respective statistical results |
7-8 |
Results and Discussion |
241 |
We have added Table 1 |
4. Audiological evaluation |
4 |
Results |
124-127 |
I have clarified the audiological results: “After the surgical procedures we observed an improvement of the clinical, functional data and also of the QoL of the operative patients. In particular, there was no recurrence of cholesteatomatous pathology at T1 in all cases. To eliminate otorrhea in only 6/52 patients (4 CWDT, 1 CWUT, 0 U-MPL, 1 O-MPL) we had to prolong the use of ear drops for a longer period than that usually foreseen by the usual follow-up medications (on average one month). Less brilliant results were obtained in the objective evaluation of hearing. Pure tone audiometry showed at T0 an auditory PTA was 55 dB in the overall population (PTA was 62 dB in patients operated with the open technique and 45 dB in those subjected to the closed technique). At T1 the means have had an improvement but not excellent (PTA in the general population was 35 dB; PTA was 45 dB and 20 dB in the open and closed techniques, respectively).” |
5. Why not study the group of patients with cholesteatoma in this study ? |
8-9 |
Discussion |
191-201 |
In the study by Maile et al. there are no statistically significant differences in QoL based on the presence or absence of cholesteatoma, although they used a generic questionnaire as the Glasgow Benefit Inventory (GBI) and not a disease-specific like COMOT-15 [23]. Likely, Weiss et al. using the ZCMEI-21 questionnaire, suggest that the presence of cholesteatoma is not decisive for the assessment of QoL, but the real factor influencing postoperative QoL is the air conduction threshold [24]….. ….In our experience most patients improved their QoL after surgery independently for the presence or absence of cholesteatoma according to previous data [23,24]. |
6. Were the 14-15 questions analysed |
|
|
|
Yes. They have been considered in the overall score. |

Round 2
Author Response
1) Reviewer 1 - Major Comments
Point-by-point responses |
Page |
Paragraph |
Line details |
|
1. Improve the parts of manuscripts … |
5 |
Material and methods |
111-112 |
The Table 1 stratifies the type of surgery according to age. |
2. Improve the parts of manuscripts … |
11 |
Discussion |
227-234 |
The latter datum (i.e., the greater satisfaction found in closed techniques) may appear in contrast with the other datum (which highlights a greater post-operative satisfaction in older patients fig. 2) since closed tympanoplasty has been preferentially performed in young patients as reported in Table 1. However, this apparent contradiction can be explained by taking into account two elements: 1) the closed techniques included not only the 7 CWUT but also 14 MPL in which older patients were also represented. 2) older patients generally started from a more complicated pre-operative condition for which even partial results led to a good degree of satisfaction. The limitations of this research are the monocentric retrospective observational design of the study, small sample size, and the exclusion of patients underwent revision surgery [23].
|
1) Reviewer 1 - Minor Comments
Point-by-point responses |
Page |
Paragraph |
Line details |
|
1.Change into … |
1 |
Summary |
38-40 |
After surgery, we observed an improvement of QoL in 84.6% of the population. The COMOT-15 overall score, ear symptoms and hearing subscores showed significantly better ratings after surgery in the whole analyzed group. However the separate analysis of patients operated with open techniques and closed techniques showed a significant improvement in ear symptoms subscore in both groups and a significant improvement in hearing subscore and mental health subscore only in patients operated with closed techniques. |
2. We have inserted the English version as well … |
4 |
Material and Methods |
95-98 |
Fig. 1 The Italian and English version of COMOT-15.
|
3. Remove - ) … |
4 |
Material and Methods |
100 |
….questions from 10 to 13.. |
4. Improve the parts of manuscripts … |
5 |
Material and methods |
111-112 |
The Table 1 stratifies the type of surgery according to age. |
5. Remove “to” |
4 |
Material and methods |
114 |
….remodelled (in 12/31 patients underwent CWDT and in 3/7 CWUT),… |
6. Change “operative” in “operated” |
5 |
Results |
124 |
of the operated patients |
7. Change “cholesteatomatous pathology” into “cholesteatoma” |
5 |
Results |
124-125 |
there was no recurrence of cholesteatoma at T1… |
8. Improve linguistic style |
5 |
Results |
127-131 |
To eliminate otorrhea in only 6/52 patients (4 CWDT, 1 CWUT, 0 U-MPL, 1 O-MPL) we have prolonged the use of ear drops as compared to the usual follow-up medications (on average one month). |
9. Change into…. |
7 |
Results |
164 |
The obtained results are summarized in Table 2.
|
10. Change into..
Explain the used abbreviations.. |
8-9 |
Results |
173 |
…Table 2: The results of COMOT-15 and its subscales for the whole group and analyzed subgroups separately…
The abbreviation are explained in text ..
|
11. Improve linguistic style |
9 |
Discussion |
184-185 |
…questionnaire before and after surgery, calculated a minimal clinically important difference (MCID) =5.3 (SD 12.0) above which the intervention can be considered satisfactory for the patient [21].
|
12. Improve the parts of manuscripts … |
11 |
Discussion |
227-234 |
The latter datum (i.e., the greater satisfaction found in closed techniques) may appear in contrast with the other datum (which highlights a greater post-operative satisfaction in older patients fig. 2) since closed tympanoplasty has been preferentially performed in young patients as reported in Table 1. However, this apparent contradiction can be explained by taking into account two elements: 1) the closed techniques included not only the 7 CWUT but also 14 MPL in which older patients were also represented. 2) older patients generally started from a more complicated pre-operative condition for which even partial results led to a good degree of satisfaction. The limitations of this research are the monocentric retrospective observational design of the study, small sample size, and the exclusion of patients underwent revision surgery [23].
|
13. Add “who” |
10 |
Conclusions |
244 |
in patients who underwent surgery |

Reviewer 2 Report
Material and Methods
Dear Authors
Some menor revision is necessary
Material and Methods:Figure 1 is not clear. It is difficult to read the information.
Discussion:It is important to comment about results related to PTA (hearing loss).
Author Response
3) Reviewer 2
1. We have changed Fig.1 |
3
|
Material and Methods
|
95-96
|
We have changed Fig.1
|
2. It is important to comment about results related to PTA (hearing loss). |
10 |
Discussion |
215-217 |
In fact, as regarding hearing loss, we obtained an improvement of an average of 20 dB in the overall population (55 dB at T0 vs 35 dB at T1). This improvement was slightly better in closed techniques (45 dB at T0 vs 20 dB at T1) respect to the open techniques (62 dB at T0 vs 45 dB at T1). |
